# Molecular Classification of Genes Associated with Hypoxic Lipid Metabolism in Pancreatic Cancer

**DOI:** 10.3390/biom12101533

**Published:** 2022-10-21

**Authors:** Yaning Li, Xinyue Liang, Gang Che, Yutong Chen, Lisi Luo, Kecheng Liu, Rongzhi Xie, Linjuan Zeng

**Affiliations:** Department of Abdominal Oncology, The Cancer Center of the Fifth Affiliated Hospital, Sun Yat-sen University, Zhuhai 519000, China

**Keywords:** pancreatic cancer, HIF-1α, hypoxia lipid metabolism, sphingolipid

## Abstract

Abnormal lipid metabolism often occurs under hypoxic microenvironment, which is an important energy supplement for cancer cell proliferation and metastasis. We aimed to explore the lipid metabolism characteristics and gene expression features of pancreatic ductal adenocarcinoma (PDAC) related to hypoxia and identify biomarkers for molecular classification based on hypoxic lipid metabolism that are evaluable for PDAC prognosis and therapy. The multiple datasets were analyzed integratively, including corresponding clinical information of samples. PDAC possesses a distinct metabolic profile and oxygen level compared with normal pancreatic tissues, according to the bioinformatics methods. In addition, a study on untargeted metabolomics using Ultra Performance Liquid Chromatography Tandem Mass Spectrometry(UPLC-MS) revealed lipid metabolites differences affected by oxygen. Analysis of PDAC gene expression profiling in The Cancer Genome Atlas (TCGA) revealed that the sphingolipid process correlates closely with HIF1α. According to the characters of HIF-1 and sphingolipid, samples can be clustered into three subgroups using non-negative matrix factorization clustering. In cluster2, patients had an increased survival time. Relatively high MUC16 mutation arises in cluster2 and may positively influence the cancer survival rates. This study explored the expression pattern of lipid metabolism under hypoxia microenvironment in PDAC. On the basis of metabolic signatures, we identified the prognosis subtypes linking lipid metabolism to hypoxia. The classifications may be conducive to developing personalized treatment programs targeting metabolic profiles.

## 1. Introduction

Pancreatic cancer remains a stable incidence that is associated with the prevalence of obesity, diabetes, and alcohol consumption [1]. In recent decades, the number of pancreatic cancer cases was almost the same as the number of deaths. Intensive chemotherapy, such as FOLFIRINOX—the combination of fluorouracil, leucovorin, irinotecan, and oxaliplatin—has resulted in longer overall survival than gemcitabine as first-line treatment in metastatic pancreatic cancer [2]. However, the five-year survival of pancreatic ductal adenocarcinoma (PDAC) in advanced stages is dismal and has not been improved significantly by current therapeutic options. In order to break through the bottleneck of chemotherapy, genomics and proteomics may potentially guide the development of new therapies [3].

Oxygen is rather essential to drive bioenergetics for energy metabolism in cells. We understand the cellular changes in O_2_ levels through the discovery of hypoxia-inducible factors (HIFs) and the regulation by the von Hippel–Lindau (VHL) tumor suppressor protein [G] (pVHL) and prolyl hydroxylases (PHD1-3 or EGLN1-3), members of the α-ketoglutarate dioxygenase superfamily [4,5]. Changes in the decrease of O_2_ levels occur in the epigenome, noncoding RNAs, the metabolome, biochemical reactions, and diverse homeostatic measures of survival under hypoxic stress [6]. Adaptation to oxygen starvation exists in both primary and metastatic neoplasms. Hypoxia exists in a large number of serious tumor types, and it elicits a wide variety of adaptive changes. In tumors, the high rate of proliferation and metastasis, abnormal tumor-associated angiogenesis, and drug resistance are related to the occurrence of hypoxia [7]. The cancer cells show the important signs of deregulation of lipid metabolism. Lipid can function as components of cellular membranes when it is also used as mediators of cancer-relevant phenotypes on transformation promotion and growth [8]. Many drugs targeting lipid metabolism have been developed as a therapeutic strategy for cancer and the inhibition of certain vital enzymes involved in lipid metabolism could inhibit progression and metastasis [9]. Related research has proved that there is significant disorder of lipid metabolism related to the Hypoxia inducible factor (HIF) in clear cell renal cell carcinoma (ccRCC) [10]. Hypoxia, as well as lipid disorder, is vital alternation of the tumor environment. Recent research has confirmed that HIF-1 regulated lipid metabolic reprogramming and influenced lipid accumulation and lipolysis in the progression of cancer [11,12]. Therefore, a method of targeting lipid metabolic alterations by HIF-1 has great potential for cancer therapy. A metabolic view from metabolomics could provide a new angle on familiar subjects—cancers.

In this study, we analyzed metabolomics along with multiple public datasets by bioinformatic tools to explore data and the clinical features of PDAC with a focus on the hypoxia lipid metabolism. We identified metabolic subtypes by the gene expression characteristics and possible molecular mechanisms according to the combination of hypoxia and lipid metabolism. Our findings suggest that there exist subgroups based on hypoxia and sphingolipid which are highly related to PDAC gene mutation and patient survival.

## 2. Materials and Methods

### 2.1. Untargeted Metabolic Analysis and Data Preprocessing

The human PDAC cell lines PANC-1 and CFPAC-1 were obtained from ATCC and maintained in DMEM or IMDM medium (Invitrogen, Carlsbad, CA, USA) containing 10% fetal bovine serum (Invitrogen) and 1% penicillin/streptomycin (Invitrogen) at 37  °C with 5% CO_2_. We chose a chemical hypoxia model -cobalt chloride [13]. Some studies have proven that it has the similar effect as the treatment of 1% O_2_ in cellular metabolism [14,15]. Approximately 8 × 10^6^ cells were seeded on the 100 mm plates (Jet Biofil, Guangzhou, China) with or without 200 μM CoCl_2_ (Sigma, Germany) for 24 h. Samples collected with 1 mL ultra-pure water included approximately 10^7^ cells with different culture conditions and were maintained and labeled. Quality controls (QCs) were run for quality assessment. Ultra Performance Liquid Chromatography Tandem Mass Spectrometry (LC-MS) peaks were identified according to mass (m/z) and retention time (RT). We used MetaboQuest (http://omicscraft.com/tools/, accessed on 20 April 2021) that can search mass values against major compound databases, including HMDB [16], METLIN [17], KEGG [18], MMCD [19], and LIPID MAPS [20] to assign putative IDs to the analytes detected by LC-MS. Metabolites with |log2-fold change (FC)| ≥ 1 were considered as differential metabolites. MetaboAnalyst (www.metaboanalyst.ca, accessed on 2 May 2021) [21] was used to upload metabolites lists obtained from previous statistical analysis for metabolite set enrichment analysis by pathway-associated metabolite sets (SMPDB [22] or KEGG).

### 2.2. Acquisition of PDAC Datasets

The RNA-seq data of GSE28735 and GSE62452 was downloaded from the Gene Expression Omnibus (GEO, https://www.ncbi.nlm.nih.gov/geo/, accessed on 15 March 2021) database. The following four RNA-seq datasets were merged into one metadata set. The expression profiles, corresponding clinical information, and somatic mutation data of PDAC were obtained from The Cancer Genome Atlas(TCGA) database (https://portal.gdc.cancer.gov/, accessed on 17 May 2021). Gene expression data, somatic mutation data, and clinical data were retrieved from the International Cancer Genome Consortium (ICGC) portal (https://dcc.icgc.org/projects, PACA-CA, PACA-AU, accessed on 16 May 2021). The mRNA expressional profiles and clinical information of PDAC cohort GSE71729 (16 May 2021) from GEO were downloaded. Samples were filtered to exclude any samples labeled as cell lines, xenografts, or normal. A total of 563 samples were ultimately included in the cluster. All the cohort mRNA matrixes were normalized by Z-score.

### 2.3. DEGs Analysis in GEO Datasets

If there were multiple probes annotated to the same gene, we chose the probe with the maximum expression level to represent the expression level of this gene. The data were processed by log2 transformation for analysis. Statistical software R (version 4.1.2, https://www.r-project.org/, accessed on 2 April 2022) and packages of Bioconductor (http://www.bioconductor.org/, accessed on 2 April 2022) [23] were downloaded to conduct significance analysis of different expression genes (DEGs) between PDAC samples and normal samples. Limma package of Bioconductor was used to select significant DEGs [24], where the standard was FDR < 0.05 and |log2-fold change (FC)| ≥ 1. Employing DAVID(https://david.ncifcrf.gov/home.jsp, accessed on 17 May 2022) [25], we explored significant DEGs by gene ID conversion and Gene ontology (GO) term enrichment analysis, including biological process, cellular component and molecular function, and KEGG pathway enrichment analysis. *p*-values were adjusted for multiple testing using false discovery rate(FDR) method.

### 2.4. Identification of Subtype-Relevant Pathways

We calculated mRNA expression of genes in mRNA expression by Spearman’s correlation in the TCGA dataset. It was considered as positive correlation with HIF1α when Spearman’s correlation was >0 and the q-value was less than 0.05. We integrated and processed the expression data frame and clinical data in the TCGA dataset and divided samples into two groups (low, high) according to the median values of each gene mRNA expression. Survival curves were plotted using the Kaplan–Meier (K-M) method by means of the survival package. The analysis of overall survival (OS) was considered statistically significant when it showed that *p* < 0.05. Gene Expression Profiling Interactive Analysis 2 (GEPIA2, http://gepia2.cancer-pku.cn, accessed on 19 May 2022) [26] merged TCGA normal and Genotype-Tissue Expression(GTEx) sample data and transformed expression data by log 2(TPM + 1). Parameter was set as |log2FC| Cutoff < 1 and q-value < 0.05.

### 2.5. Metabolic Subgroup Classification and Subgroup Analysis

Genesets “KEGG_HIF-1_signaling_pathway” (hsa04066) and “GOBP_SPHINGOLIPID_METABOLIC_PROCESS” (GO:0006665) were used as hypoxia lipid metabolism genes. Consensus clustering was performed using ConsensusClusterPlus [27]. Euclidean distances were used as the distance metric. The clustering number was set as 10, and the consistency matrix and consistency cumulative distribution function were calculated to determine the best classification. Using the “GenVisR” package [28], somatic mutation data were visualized. With consensus clustering, we classified the dataset according to preexisting PDAC subtypes Bailey [29] and Moffitt [30]. The Bailey subtype was based on DEGs results from the publication result [24], which were filtered for genes with FDR <0.05 and |log2-fold change| >1, resulting in 214 ADEX, 1182 squamous, 164 pancreatic progenitor, and 297 immunogenic genes. The Moffitt subtype was based on the 50 gene signatures in publication result [25].

### 2.6. Quantitative Real-Time PCR (RT-PCR)

We used an RNA Quick Purification kit(RN001, ESscience, Shanghai, China) to extract total RNA. The PrimeScript™ RT Master Mix reagent kit (TaKaRa, Beijing, China) was used for mRNA reverse transcription according to the manufacturer’s instructions. Quantitative RT-PCR was performed using an AriaMx Real-Time PCR System (Agilent, Palo alto, CA, USA) with TB Green^®^ Premix Ex Taq™ (TaKaRa). Primer sequences (Sangon Biotech, Shanghai, China) for quantitative RT-PCR are shown in Appendix A. By the ΔΔCt method, the relative expression levels of the target genes were analyzed. We performed the experiments in triplicate and repeated at least three times.

### 2.7. Statistical Analysis

The statistical analysis was performed with GraphPad Prism9 statistical software and R 4.1.2 (https://www.r-project.org/, accessed on 2 April 2022). The comparisons of two-sample data were performed using t-tests. Kaplan–Meier survival analysis was used to compare the differences in OS between different groups. *p* or FDR < 0.05 was considered statistically significant.

## 3. Results

### 3.1. Subsection

#### 3.1.1. Identification of Oxygen Change and Lipid Catabolism Changes in PDAC

In order to identify gene expression alterations, we separately screened the differentially expressed genes (DEGs) in GSE28735 and GSE62452 with pairs of normal and PDAC tissues. The results of DEGs were visualized by the volcano plot (Figure 1a). Between the two data sets, 302 DEGs were overlapping and further examined for biological function using GO functional enrichment and KEGG pathway analyses (Appendix A). Among the enriched results, it showed that DEGs were related with hypoxia and lipid metabolism, such as lipid catabolic process (GO:0016042), response to hypoxia (GO:0001666), triglyceride lipase activity (GO:0004806), and lipid digestion (GO:0044241) (Figure 1b). In addition, KEGG pathway analysis showed that the related pathway included fat digestion and absorption (KEGG:hsa04975) and Glycerolipid metabolism (KEGG:hsa00561) (Figure 1c). The results showed that PDAC exhibits oxygen change and some alterations of lipid metabolism, mostly on catabolism.

#### 3.1.2. LC-MS-Based Metabolomic Analysis and the Metabolomic Differences Analysis of Normoxic and Hypoxic Culture

To further understand the metabolites in the state of cells, we selected two types of PDAC cell lines and investigated. First, we constructed an in vitro hypoxia model in two cell lines, PANC-1 and CFPAC-1, using CoCl_2_. The sufficient induction of hypoxia was confirmed by increasing expression of HIF-1α protein (Appendix A). We performed metabolomic analysis of samples from cultured cells under normal or simulative hypoxia condition, respectively, using LC-MS. As shown in Figure 2a, a total number of 266 identified metabolites were analyzed. Among them, 164 metabolites were identified as enriched, while 102 types were shown as depleted. The annotation of metabolites number was shown in Appendix A. The heat map implies that in both cell lines, a number of metabolites have strong changes that resulted in metabolic variation. (Figure 2b). In hypoxic, metabolites such as Phosphatidylcholine and Ethanolamine phosphate were overexpressed at the same time. They could interfere with the Glycerophospholipid metabolism. Cancer cells with high proliferation need a large amount of glycerophospholipids, particularly for membrane production. In two types of cell lines, the substance including L-histidine, Carnosine, and L-Glutamate involved in histidine metabolism was also down-regulated under hypoxia. The intake and metabolism of histidine was shown to influence the sensitivity to methotrexate in cancer cells [31]. SMPDB (The Small Molecule Pathway Database) is a database containing more than 30,000 small molecule pathways found in humans only, and the majority of these pathways are not found in other pathway databases. It is designed specifically for pathway discovery in metabolomics, transcriptomics, proteomics, and systems biology. The metabolites such as adenosine monophosphate, L-Carnitine, glycerophosphocholine, and 1,2-dipalmitoyl-sn-glycero-3-PC have changed, leading to the suppressed lipid metabolic effect such as beta oxidation of very long chain fatty acids and oxidation of branched chain fatty acids in SMPDB (Figure 2c). This was in good agreement with the results of lipid catabolic changes in the GEO datasets above. On the other hand, some substances, such as L-Serine and Choline, remained unchanged; they are the participants in Glycine, serine, and threonine metabolism. This might be because many factors could influence Glycine, serine, and threonine metabolism, such as obesity [32] and diabetes [33] as well as hypoxia. We also noticed that some metabolite changes are not consistent between PANC-1 and CFPAC-1; this might be attributed to the different origin of the cell lines. PANC-1 is an epithelioid cell line started from a human pancreatic ductal carcinoma, while CFPAC-1 is a cell line with the characteristics of pancreatic duct cells originated from a patient with a tumor in the head of the pancreas and cystic fibrosis (CF). It expresses the CF gene and manifests the most common CF mutation. The different metabolic changes between the two cells strongly suggest that patients with pancreatic cancer may also have different metabolic patterns.

#### 3.1.3. Transcriptomics Data Analysis Focus on HIF-1α

To get more precise and deeper analysis results, we downloaded the data of PDAC patients from TCGA, which possesses large-scale patient samples and complete data in multiple kinds of cancers. We aimed to investigate changes by genes affected by hypoxic environment, and HIF-1α was thought to be the core of the hypoxia. The gene expression profiles of the PDAC dataset obtained from TCGA were analyzed, and a total of 5874 genes were statistically significant (q-value < 0.05) and positive correlated (Spearman’s correlation > 0) with HIF-1α in 186 patients. The top 20 ranked genes are visualized in Figure 3a. Through the Kyoto Encyclopedia of Genes and Genomes (KEGG) database, the above genes were further analyzed by enrichment pathways analysis. Finally, 122 KEGG pathways were enriched. It was significantly shown that staphylococcus aureus infection attained a higher enrichment score. Under hypoxia, the capacity of neutrophils to kill bacteria could be impaired [34]. The result might explain why staphylococcus aureus infection was markedly affected, and it was regarded as a significant cause of morbidity and mortality in cancer patients. In results, quite a number of genes were hit in the pathway of focal adhesion. One of the critical steps in cancer progression cell adhesion seems to be related to the changes of HIF-1α. In addition, there were some lipid-related pathways. Sphingolipids participate in a wide variety of biological mechanisms as bioactive lipids; here is the evidence that imbalances in sphingolipid metabolism exist in many tumors. It plays a role in cell death, survival, and therapy resistance in cancer. For example, ceramide, as a member of sphingolipid and a regulator of apoptosis, is one of the molecular obstacles that inhibit cancerous cells from achieving necessary proliferation. In recent years, modulation targeting the sphingolipid metabolic pathways has been on the leading edge of drug discovery for cancer therapeutics. Notably, sphingolipid signaling pathway was included (Figure 3b). Using the Kaplan–Meier survival analysis [35], batch filtering of positively correlated gene results displayed 504 genes that have significant association with the OS of PDAC patients in the TCGA dataset. Survival time differed in patients with the relatively high versus low expression of protein of these genes. These were all factors resulting in the patients’ poor survival in PDAC. We evaluated the expression of the genes above in normal and tumor samples of the TCGA and GTEx databases. The mRNA expression levels of genes in PDAC cancers were analyzed by GEPIA2, and 252 genes were significant. Using multiple filter genes, a KEGG item—Sphingolipid signaling pathway—showed significantly differential enrichment, and the overlapping genes were GNAI3, KRAS, NRAS, MAPK1, SGMS2, and PPP2R5E. The mRNA expression level of crossover genes and Kaplan–Meier survival analysis (GNAI3, KRAS, NRAS, MAPK1, SGMS2, and PPP2R5E) in this metabolic pathway are shown in Appendix A. Because the KRAS and NRAS mutation are thought to be the overwhelming inner characteristics of PDAC [36], we detected the mRNA level of four other genes in two PDAC lines and found that hypoxic culture markedly increased expression of GNAI3, MAPK1, SGMS2, and PPP2R5E, which are involved in the process of Sphingolipid signaling pathway. (Figure 3c).

#### 3.1.4. PDAC Subtypes Identified by HIF-1 and Sphingolipid Genes

We integrated RNA-seq data from multiple PDAC datasets including TCGA PAAD, GSE71729, ICGC PACA-CA, and PACA-AU. A total of 563 patients remained in the sample after excluding non-PDAC samples and samples without clinically informative features (in detail: 176 of TCGA, 125 of GSE71729, 182 of PACA-CA, and 80 of PACA-AU). We obtained genes belonging to the KEGG gene set HIF-1 signaling pathway (hsa04066) (n = 109), and the GO gene set SPHINGOLIPID METABOLIC PROCESS (GO:0006665) (n = 162) was used for consensus cluster analysis, which is a highly effective technique in biological research. The approach combines data from different experiments, increases the credibility in the common features of all the datasets, and reveals the important characteristics among them. We chose K = 3 as the optimal choice by the corresponding cumulative distribution function (CDF) curve and the delta area plot (Appendix A). Consensus clustering method was applied on the merged dataset to cluster the PDACs into three subtypes (Figure 4a). The cases of group cluster2 in merging data was 32.5% (183/563), followed by cluster1 10.48% (59/563), and cluster3 57.02% (321/563). Principal component analysis(PCA) (Figure 4b) indicated that groups could successfully discriminate and classify samples according to the characteristic subtype classification. Further, we investigated the three subgroups’ prognoses. The survival curve indicated that the samples in cluster1 invariably had shorter survival time, in days, while cluster2 predicted the longest survival time. Overall, the survival results in cluster3 that are similar to the cluster1 were poorer (Figure 4c). The results revealed that the hypoxia lipid metabolic subtypes had existed in PDAC, and they were accompanied by a different prognoses.

#### 3.1.5. Distribution of Mutation Characteristics and Association with Other PDAC Subtypes

Expression levels of HIF-1 and sphingolipid genes among the metabolic subgroups are visualized in Appendix A. The heatmap indicated that there is a subtle difference among mRNA expression in the clusters. Further, we analyzed the alternation of genes participating in HIF-1 signaling pathway among different groups. The cluster3 had many more changes on related genes than cluster2 (Figure 5a). BCL-2 protein could control cell death and significantly changed in cluster3. RELA, as a critical transcriptional factor for response to hypoxia, increased in cluster1 and 3 by comparison with cluster2. ELOVL1, SMPD3, and DEGS1 were involved in sphingolipid metabolic process and achieved the rise in cluster1 and 3. Gene expression analysis revealed increased expression of HIF1α-associated genes P4HA1 in cluster3 and increased expression of the sphingolipid signaling genes SPHK1 and SPHK2 in cluster1 (Appendix A). To clarify the clinical signature difference among the three clusters, the results demonstrated that there was no significant difference based on age or gender (Appendix A). To search for affected pathways associated with mRNA expression in the clusters, we performed a comprehensive analysis, including differentially expressed analysis and functional enrichment. Filtered genes revealed significant difference of function. Pathways enriched among cluster1 correlated genes included ubiquitin-mediated proteolysis, autophagy, TGFβ signaling and lipid, and atherosclerosis. Chemokine signaling pathway, cholinergic synapse, and calcium signaling pathways were remarkably affected in cluster2. Ras signaling pathway, MAPK signaling pathway, and Rap1 signaling pathway changed in samples divided into cluster3 (Appendix A). In the somatic alterations, the most common driver genes (KRAS, CDKN2A, TP53, and SMAD4) were still in the top position. There was a remarkably higher variation in MUC16 mutation in cluster2 than in the cluster1 and cluster3 subgroups (Figure 5b). Previous studies indicated that the mutation number of MUC16 was closely correlated with tumor mutational burden (TMB), and the high mutation number of MUC16 was correlated with better overall survival (OS) [37,38]. There exists PDAC gene expression subtypes associated with survival in previous studies. The basal-like (Moffitt [25]) and squamous (Bailey [24]) groups have the poor outcomes. To investigate the relationship between our classification and previous subtypes, we divided samples into the various subtypes and analyzed the composition of each cluster (Figure 5c). The cluster1 and 3, with worse outcomes, included mainly basal-like cases (59.32%; 82.86%) and contained squamous (38.98%; 34.57%), respectively. The majority of cluster2 with longer survival time were classical samples (63.93%) and had fewer samples in the basal-like (36.1%) and squamous groups (4.37%). The results show that pathways related to hypoxia lipid metabolism can help us recognize the PDAC subtypes with different prognoses.

## 4. Discussion

Though there are improvements in survival of PDAC patients under the first-line and second-line palliative therapies and adjuvant treatment, the overall 5-year survival for pancreatic cancer has still changed little. The urgency for the development of personalized treatment has created the focus on the relevant tumor subtypes [39]. The molecular heterogeneity in PDAC leads to the diverse types accompanied with the difference of gene expression and structural variations [24,25]. For example, PDAC patients with BRAC1/2 mutation are often sensitive to cisplatin therapy and benefit from subsequently maintenance therapy using Olaparib [40]. Although BRAC1/2 mutation occurs only in 4–7% of PDAC patients, it forges a path for individual therapy based on gene features. The more we know about the gene profile characteristics of PDAC, the more chance for us to translate the variation information into clinical practice for outcome prognostication and treatment choice to improve the survival benefit. Our results show that, based on the oxygen regulation and sphingolipid metabolism, the gene expressions were significantly different and led to the patient outcome discrepancy.

Hypoxia facilitates pathways—such as angiogenesis, growth-factor signaling, genetic instability, apoptosis, invasion, and metastasis—and increases resistance to radiation therapy and chemotherapy [41]. When the oxygen level is reduced, cells respond through hypoxia-inducible transcription factor 1 (HIF-1). HIF-1 consists of the hypoxic response factor HIF-1α and the constitutively expressed aryl hydrocarbon receptor nuclear translocator (ARNT) (also known as HIF-1β). In the hypoxia status, HIF-1 binds to hypoxia-response elements (HREs), activating the downstream genes. The activation of hypoxia-inducible factor 1 (HIF1) up-regulated the sterol regulatory element-binding protein (SREBP)-1, which can impact FAS by transcriptional regulation [42]. GLS1 mRNA and protein expression are associated with hypoxia-inducible factor and are involved in tumor growth and metastasis in colorectal cancer [43]. Lipids are important elements in many overlapping oncogenic signaling pathways. Status of hypoxia may influence the anti-tumor-immunity process and lead to the loss of immunotherapy response in cancers [44]. Therefore, the treatment targeting hypoxia to overcome the resistance of tumor cells and immunotherapy is a highly potential research area. To target hypoxic tumor cells, approaches including hypoxia-activated prodrugs, gene therapy, specific targeting of HIFs, and targeting pathways related with hypoxia have been researched widely [45].

Sphingolipids are the structural components of cell membranes and also regulate growth, proliferation, migration, invasion, and metastasis by signaling functions; additionally, the sphingolipid metabolism process generates resistance to chemotherapy and radiotherapy. Tumors exhibit increased ceramide metabolism mainly by increased activities of materials such as glucosyl-ceramide synthase (GCS) and increases the generation of sphingolipids [46]. Therapeutic targeting of sphingolipids chemicals, such as FTY720 targeting sphingolipid signaling, has demonstrated potential function in cancer therapy. Studies indicate antiproliferative effect, apoptosis-inducing ability, and drugs of additive effect of FTY720 in diverse cancer cells like pancreatic cancer [47,48].

In this study, we utilized multiple bioinformation analyses to identify the impact of gene-associated hypoxia lipid metabolism in PDAC. PDAC transcriptome analysis and untargeted metabolic analysis demonstrated that a low level of HIF-1α protein is correlated with the lipid metabolic process. In addition, on the basis of the signature of HIF-1 and sphingolipid, we classified samples into three different clusters with different molecular characteristics and survival outcomes. The results demonstrated that cluster2 predicted long survival time, and the MUC16 mutation is high. MUC16 (previously known as CA125) is a type I transmembrane mucin protein with a C-terminal domain, a tandem repeat region, and an extracellular N-terminal section. Several studies have shown that MUC16 was found to be one of the top frequently mutated genes. Its overexpression affects the growth and metastasis function [49,50]. In addition, the group of patients with MUC16 mutation had better survival outcomes in gastric cancer [26,27]. In cluster3, samples expressed relatively higher gene expression of P4HA1 and had short overall survival. The role of P4HA1 in PDAC was revealed, which can regulate HIF1α activity and led to high proliferation, chemoresistance, and cancer cell stemness [51]. Targeting P4HA1 may be a potential therapeutic approach for PDAC. By observation, we note that the levels of SPHK1 and SPHK2—which are key enzymes in the phosphorylation of sphingosine to sphingosine 1-phosphate (S1P)—were highest in the cluster1. The SPHK2 inhibitors ABC294640 and SK1-I (a competitive inhibitor of SPHK1) have proved their clinical value and can suppress the growth of tumors via multiple mechanisms [52,53].

Our research proves that PDAC has the striking disorder in lipid metabolism that may be regulated by the hypoxia environment. Metabolism-associated molecular classification relying on hypoxia lipid metabolism genes has decent concurrence with previous molecular subtypes in patients’ prognosis. With this new viewpoint, we investigate the classification focus on lipid metabolism in PDAC. Doubtless, with the help of the metabolism difference, doctors can divide patients by priority and estimate the prognosis and exploit the specific and precise therapeutic strategies targeting multiple metabolic dependencies.

## 5. Conclusions

By enriched analysis in different expression genes significantly, we find abnormal lipid metabolisms and oxygen changes exist in PDAC. Untargeted metabolomics in two types of PDAC cells clearly show HIF-1α protein-associated metabolites. By identifying metabolites likely perturbed in hypoxia, the analysis included many lipid metabolism pathways. The identification of the genes positively with HIF-1α in the TCGA PDAC data resulted from our multiple analyses. The sphingolipid signaling pathway was enriched using the genes screened. On the basis of genes of HIF-1 pathway and sphingolipid process, three subclasses were identified. Subgroups are found to correlate to different patient survival. Patients in cluster2 had a significant survival advantage over those in the other groups.

## Figures and Tables

**Figure 1 biomolecules-12-01533-f001:**
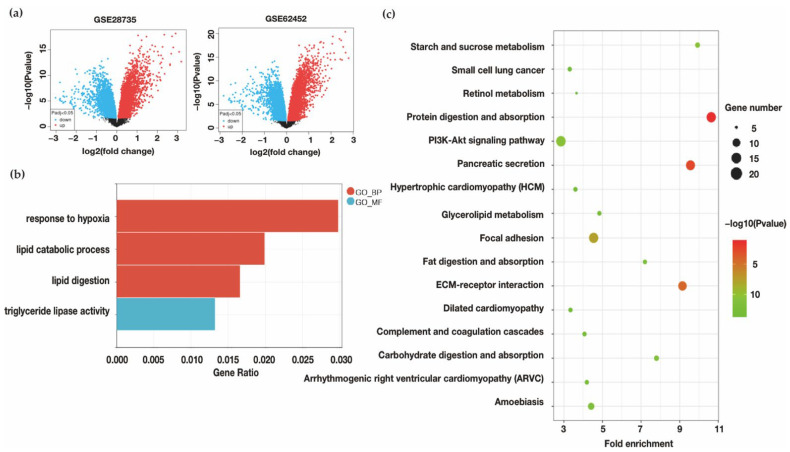
Identification of the occurrence of lipid metabolism disorder and oxygen change in PDAC datasets. (**a**) The volcano plot of analyses of DEGs in GSE28735 and GSE62452. (**b**) The GO molecular functions, biological processes, and cell components related to oxygen and lipid change. (**c**) The KEGG pathways enrichment in overlapping DEGs. (GO _BP: Gene ontology biological process; GO _MF: Gene ontology molecular function) (GSE28735: https://www.ncbi.nlm.nih.gov/geo/query/acc.cgi?acc=gse28735; GSE62452: https://www.ncbi.nlm.nih.gov/geo/query/acc.cgi?acc=gse62452, accessed on 15 March 2021).

**Figure 2 biomolecules-12-01533-f002:**
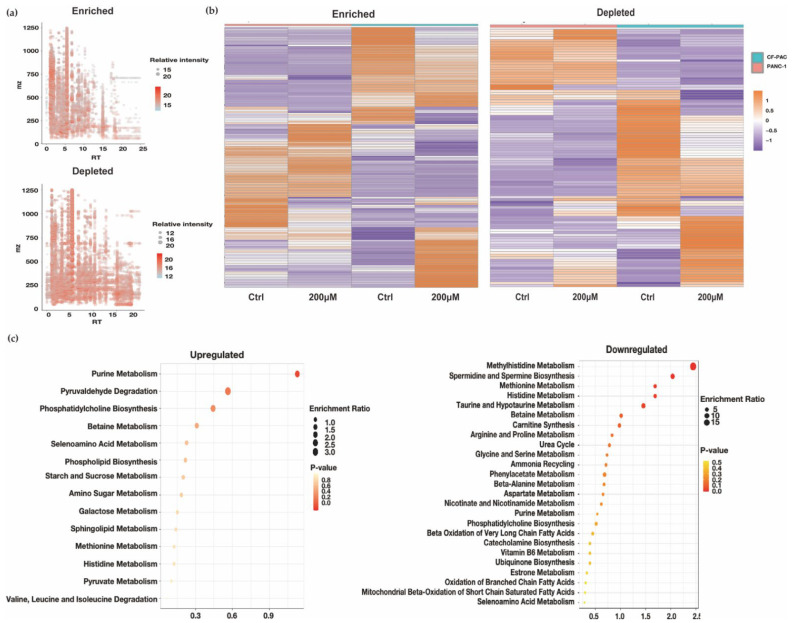
Analysis of metabolomic data of CoCl_2_ treated PDAC cell lines. (**a**) Bubble chart of all metabolite molecules detected. The abscissa is the retention time, the ordinate is the mass-to-charge ratio, the size of the point is the relative abundance of metabolites. (**b**) Heatmap of metabolites in PANC-1 and CFPAC-1 cells affected by the use of CoCl_2_. (**c**) The SMPDB pathways enriched in upregulated and downregulated metabolites.

**Figure 3 biomolecules-12-01533-f003:**
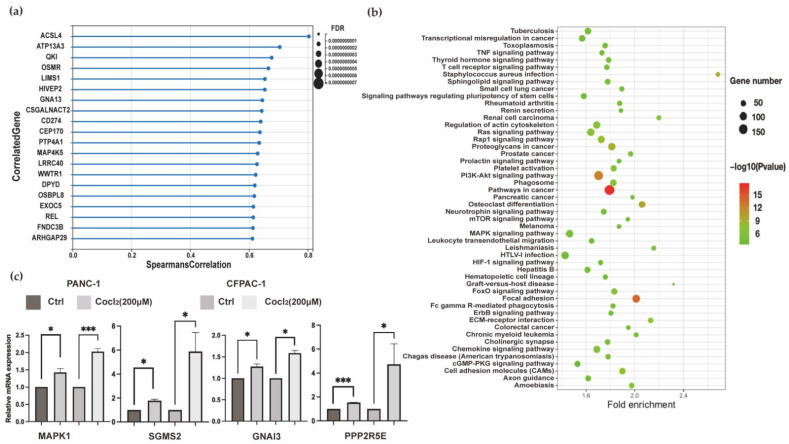
Identification of HIF-1α-related genes and biofunction by bioinformatics analysis. (**a**) Statistical chart of the top 20 genes positively related to HIF-1α. (**b**) The KEGG pathways enriched in genes selected by Spearman’s correlation coefficient analysis. (**c**) The mRNA expression of genes involving in sphingolipid signaling pathway in two cell lines. (n = 3. *, *p* < 0.05; ***, *p* < 0.001). (FDR: false discovery rate) (TCGA: https://portal.gdc.cancer.gov/, accessed on 17 May 2021).

**Figure 4 biomolecules-12-01533-f004:**
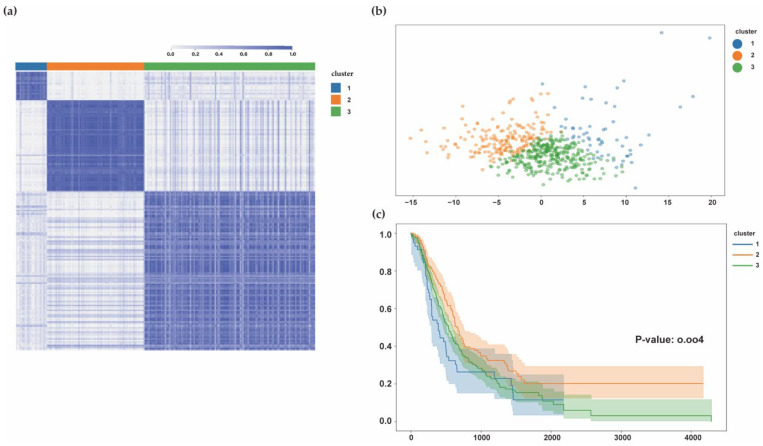
Metabolic subtypes rely on HIF-1 and sphingolipid pathways (cluster1 = 59, cluster2 = 183, cluster3 = 321). (**a**) Clustering heatmap (k = 3) of metabolism pathways in 563 samples. (**b**) PCA (Principal Component Analysis) plot of three cluster groups under NMF method. (**c**) Kaplan–Meier survival analysis of patients in multiple metabolic subgroups. (TCGA: https://portal.gdc.cancer.gov/, accessed on 17 May 2021; GSE71729: https://www.ncbi.nlm.nih.gov/geo/query/acc.cgi?acc=gse71729, accessed on 16 May 2021; ICGC: https://dcc.icgc.org/projects, accessed on 16 May 2021).

**Figure 5 biomolecules-12-01533-f005:**
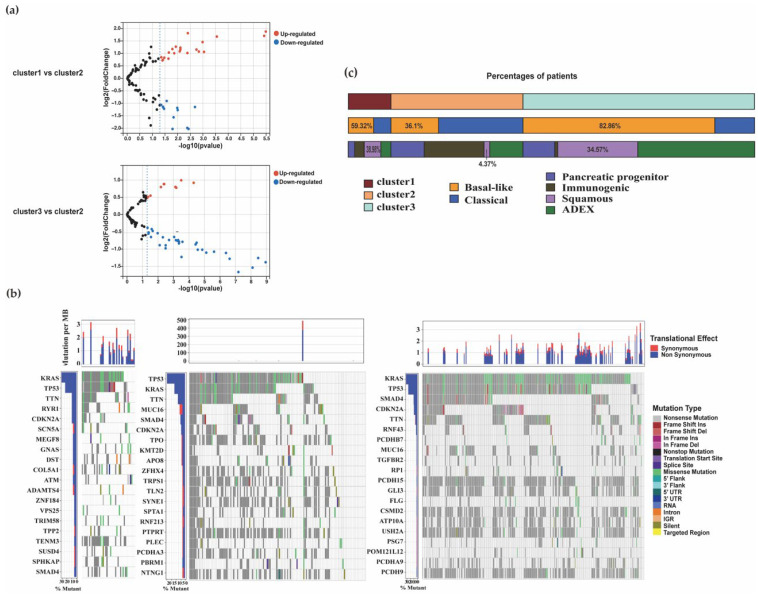
Characters of the PDAC subtypes and comparison with other prognostic subtypes. (**a**) The volcano charts of HIF-1-related genes among groups. (**b**) The somatic mutations and copy number mutations of the three clusters. (**c**) The comparison bar chart of the number of patients in PDAC subtypes with Moffitt and Bailey. (TCGA: https://portal.gdc.cancer.gov/, accessed on 17 May 2021; GSE71729: https://www.ncbi.nlm.nih.gov/geo/query/acc.cgi?acc=gse71729, accessed on 16 May 2021; ICGC: https://dcc.icgc.org/projects, accessed on 16 May 2021).

## Data Availability

The datasets used during the study are available in the Gene Expression Omnibus (GEO, https://www.ncbi.nlm.nih.gov/geo/, GSE28735 & GSE62452: 15 March 2021, GSE71729: 16 May 2021), The Cancer Genome Atlas (TCGA) database (https://portal.gdc.cancer.gov/, accessed on 17 May 2021) and the International Cancer Genome Consortium (ICGC) portal (https://dcc.icgc.org/projects, accessed on 16 May 2021). Primers used in the present study can be found in Additional file. 2. Other original data in our study are available upon request.

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
