# Peer review of "Molecular Classification of Genes Associated with Hypoxic Lipid Metabolism in Pancreatic Cancer"

_biomolecules, 2022, doi:10.3390/biom12101533_

Round 1

Reviewer 1 Report

Thank you for the opportunity to review this interesting manuscript, dealing with important findings entitled “Molecular classification of genes associated with hypoxic lipid metabolism in pancreatic cancer”. All findings are interesting, and the article includes a balanced and critical view of the findings. I am unable to find sample size and statistical analysis in some figure legends such as Figure S1a Sb. The author needs to edit all figure legends with proper statistical explanations with sample size. 

Author Response

Thanks to the editor's comments, please see the attachment.

Reviewer 2 Report

Methods line 68-70. “…approximately 107 amounts of cells…” is unclear. How many cells were seeded on what size of plates? How long were they treated? How was the cell culture supernatant collected? The amount of significant information missing form these methods is concerning.

Line 150 regarding figure 1B. How many overlapping genes were observed and used in the GO enrichment analysis? It might be useful to show this information in a Venn diagram. Were these the most enriched pathways or selected pathways.

Lines 169-170. The authors should change to enriched/depleted rather than positive mode and negative mode.

Line 178 regarding Figure 2B and 2C. The authors should define SMPDB for readers unfamiliar with this analysis. How many metabolites were used in the SMPDB analysis for Figure 2C? Was this data combined from both cell lines or just one of the cell lines? The authors should discuss those metabolites that are commonly altered. Of note, based on the heatmap, there do not appear to be that many consistently altered metabolites in both cell lines.  The authors discuss a few of the pathways but not the most significant and never discuss relative differences between the cell lines. This should be addressed.

Table 1. should be supplementary information. This table should also include the relative abundance of the metabolite in each cell line compared to the NC.

Line 192. How many patients from TCGA was analyzed? The authors should state the significance of investigating the genes which correlate with HIF1a expression.

Figure 3A. The legend states top 50 genes correlated, the text states the top 20 (line 194). There are 19 genes in the figure.

Figure 3B. The text discusses KEGG pathway analysis and sphingolipid signaling (lines 197-198). In the Figure, 3C should be labeled as 3B. It is unclear why the sphingolipid signaling term is significant compared to all the other terms based on Figure 3B.

Lines 202-205. The authors should more clearly state how they came to the genes indicated.

Figure S1B and S1C. The authors should define the red and gray groups in Figure S1B. Figure S1C should be higher resolution to aid readability.

Lines 207-208. Was NRAS and KRAS tested, and no change observed?

Line 218. The authors should cite the papers where these datasets originated.

Line 223. The authors should define consensus cluster analysis.

Lines 226-233 regarding Figure 4. The order in which the survival data is discuss is disorganized (lines 226-233). The figure 3c legend should include the number of samples per group. The 3 groups should be consistently color coded. Each panel should use a consistent color for each group (as in cluster 1 should always be red as in 4A).  

The authors should discuss more introduction regarding links between hypoxia and altered lipid metabolism. Similarly stating their rationale in introducing topics and analyses throughout the results would be extremely beneficial for readers.

Author Response

(The authors gave the same response as above.)

Reviewer 3 Report

Based on the results of various bioinformatics techniques, in multiple datasets, the authors were able to determine there is a specific cluster predicting increased survival for patients. They loosely describe previous work in the introduction stating that correlations have been seen elsewhere and that adding in metabolomics should be able to strength this correlation. The goal was to look at potential association for lipid alterations with hypoxia signatures.  

Pulling from different datasets, it is hard to get a clear picture of the story overall. In their main finding- there is a cluster group, where they picked out MUC16 as a highly mutated gene. In another cluster (3) with shorter survival time, they picked out that P4HA1 has higher expression. What these mean for PDAC and confirming these findings was not explored. There are some additional areas that could be improved for general understanding of the significance and unanswered questions. Testing these two genes found in their clusters, back in their cell lines by increasing/decreasing expression to study effects on altering growth/death would be a nice set of data to validate that any of these findings are physiological and relevant.  Currently there is no mechanism or validation of findings, so it is correlative only and doesn’t highlight the importance for PDAC. By rewriting certain aspects of the story and examining even one of these genes for altering ‘oncogenic potential’ this would strengthen the story.

Suggested Improvements:
• Please work on the flow of writing in the introduction, especially the transition from hypoxia to lipids. Having an additional reviewer/author to adjust the grammar/language used would also be useful for the abstract, introduction, and discussion.

Major Comments:

1. Sentence 1 of the results, “429 and 322 DEGs were separately screened in …. “, respectively(?).
It is hard to follow this from the start. Are you saying 429 genes were looked at in one set and 322 in the other? Why not the same genes looked at in both… and where did these arbitrary numbers come from? How many genes overlapped (line 149)? Maybe state ‘between both data sets, ### DEGs were overlapping and further examined for …’. Also differentially expressed compared to what? What are these relatively changed in comparison to? An introduction to what was done and how in the results would help.
               “To examine genes that are altered during PDAC we used two …”

2. Figure 2b, these two cell lines show drastically different results. Please explain these in more detail. It would be nice to have a few overlapping categories that are the same, those that are drastically unaltered and a discussion as to why.

3. From the results describing the data, I never take away a clear message. No data is ever validated for appropriateness or actual effects. You state that “pathways related to hypoxia lipid metabolism can help us to recognize the PDAC subtypes”, but I’m not sure I understand how genes of hypoxia or lipid metabolism are changed in your clusters. Just that they do cluster. Since this IS the finding of your paper, explain WHAT genes of hypoxia are altered between the clusters and why this could matter -or- go back to your cell lines -/+ hypoxia and see how these lipid metabolic genes in the clusters could push the cells to death or better survival etc.

Minor Comments:

1. Describe why you used PANC-1 and CFPAC-1. Why might they show very different results etc.

2. How does CoCL2 compare to hypoxia… what level of oxygen depletion is it similar too? In some contexts you say without oxygen, but really this is just a treatment not actually without oxygen.

3. Sentence on line 155-156, is an overstatement. Results suggest alterations in lipid pathways… and nothing more.

4. S1a – how long were cells treated with and what concentration? This should be in the figure legend clearly.
5. Line 168, overstatement. ‘found alterations in lipid profiles’.

6. Transitions throughout are hard to follow. Start with dataset, jump to cell lines, then jump back to dataset. How does cell line metabolomics compare to two analysis from different data sets. Please give an introduction to what you did and why and not just start with a very non descriptive intro to data (example, line 192).

7. Fig 3A and 3b are not legible, as well as many others. Please fix your figures in illustrator or similar program to be clearly visible -labels, axis, etc.

8. Marked expression of genes in line 210, what are these genes? Why do they matter?

9. “TMB’ and ‘OS’- don’t state things without defining them- line 261

10. Can’t read any of fig 5. Fix the data to be readable.

10. A lot of the enriched data is not really described throughout, just a candidate look at target areas of interest. In example, why are genes associated with staph enriched so much on fig3 c?

11. Make all data sets available in tables/files to follow.

Author Response

Thanks for the reviewer's comments.Please see the attachment.
